# Edge Computing Data Optimization for Smart Quality Management: Industry 5.0 Perspective

Bojana Bajic [1,2] , Nikola Suzic [3], Slobodan Moraca [1] , Miladin Stefanović [4] , Milos Jovicic [2] and Aleksandar Rikalovic [1,2,*]

1   Department of Industrial Engineering and Management, University of Novi Sad, 21000 Novi Sad, Serbia; bojana.bajic@uns.ac.rs (B.B.); moraca@uns.ac.rs (S.M.)
2   Institute for Artificial Intelligence Research and Developments of Serbia, 21000 Novi Sad, Serbia; milos.jovicic@ivi.ac.rs
3   Department of Industrial Engineering, University of Trento, 38123 Trento, Italy; nikola.suzic@unitn.it
4   Center for Quality, Faculty of Engineering, University of Kragujevac, 34000 Kragujevac, Serbia; miladin@kg.ac.rs
*   Correspondence: a.rikalovic@uns.ac.rs or aleksandar.rikalovic@ivi.ac.rs

**Abstract:** In the last decade, researchers have focused on digital technologies within Industry 4.0. However, it seems the Industry 4.0 hype did not fulfil industry expectations due to many implementation challenges. Today, Industry 5.0 proposes a human-centric approach to implement digital sustainable technologies for smart quality improvement. One important aspect of digital sustainability is reducing the energy consumption of digital technologies. This can be achieved through a variety of means, such as optimizing energy efficiency, and data centres power consumption. Complementing and extending features of Industry 4.0, this research develops a conceptual model to promote Industry 5.0. The aim of the model is to optimize data without losing significant information contained in big data. The model is empowered by edge computing, as the Industry 5.0 enabler, which provides timely, meaningful insights into the system, and the achievement of real-time decision-making. In this way, we aim to optimize data storage and create conditions for further power and processing resource rationalization. Additionally, the proposed model contributes to Industry 5.0 from a social aspect by considering the knowledge, not only of experienced engineers, but also of workers who work on machines. Finally, the industrial application was done through a proof-of-concept using manufacturing data from the process industry, where the amount of data was reduced by 99.73% without losing significant information contained in big data.

**Keywords:** human-cyber-physical systems (HCPS); big data analytics (BDA); Industrial Internet of Things (IIoT); smart quality management; digital sustainability; data optimization

## 1. Introduction

In the last decade, global hype has been built around creating smart, connected manufacturing systems based on Industry 4.0 and emphasizing the role of cyber-physical systems (CPS) [1,2]. In turn, the CPS environments, in which physical objects and software are closely integrated via the Industrial Internet of Things (IIoT) [3,4], improved the human–machine interaction but limited the socially sustainable aspect [5]. Moreover, the relevant literature reports the managerial implementation challenges [6]. These challenges pointed out the lack the human resources as the biggest barrier for hindering Industry 4.0 implementation. Thus, the technological transformations of Industry 4.0 did not carefully consider humans as a central point in the manufacturing environments [7]. Therefore, it seems that the technology-centered approach of Industry 4.0 has proved inadequate, since the lack of a human impact in the application of Industry 4.0 has been reported [6,8]. The new Industry 5.0 concept intends to put the human aspect back into the center of the

manufacturing processes [5,9], emphasizing the socially and digitally sustainable aspects with the aim of developing human-cyber-physical systems (HCPS) [1,10].

Understanding this need for a focus change, the Industry 5.0 emphasizes the synergy between humans and autonomous machines with social and environmental dimensions [7,11]. Specifically, the Industry 5.0 implies a transition to a digitally sustainable, human–centric and resilient industry [5,9]. This transition empowers the creation of HCPS. As part of the Industry 5.0 concept, HCPS represent a complex intelligent system that encompasses humans (i.e., engineers [12,13], workers [11,14] and data researchers)—considering social aspects, knowledge and experience, cyber systems, and physical objects connected via IIoT [15]—with the aim of achieving smart quality management [16] with 'self-aware,' 'self-prediction,' and 'self-maintenance' capabilities [4,17]. However, a number of challenges for implementing HCPS in industry remain [18].

More specifically, the human-cyber-physical systems require the generation of a massive amount of manufacturing data, which is called big data [19]. Big data in manufacturing [20] is highly distributed, structured, semi-structured, and unstructured raw data generated by multisource sensors through intercommunication within the system and externally related communication [21–23]. With the use of new digital technologies and smart analytical methods, big data provides new solutions to improve manufacturing system reliability and effectiveness [21–24]. Vast storage, power, and processing resources are required to cope with the volume of big data, making cloud and edge computing the main technologies of choice [25,26].

Both cloud and edge computing have advantages and disadvantages for HCPS. On the one hand, cloud computing offers high performance in storing, processing, and analyzing big data generated via the IIoT. Challenges to using cloud computing include high costs, high energy consumption, security issues, and long response times [27]. On the other hand, edge computing offers more security and real-time data processing, with lower costs and lower energy consumption. However, edge computing solutions lack storage and processing power [27,28] where the data optimization is required to reduce the number of data samples. To the best of our knowledge, the relevant literature does not provide a solution for HCPS that fulfils the criteria for efficient and sustainable big data analytics and optimization at low costs.

The present research aims to fill this gap with the efficient optimization of big data and further power and processing resource rationalization using edge computing technology for resilient production. Specifically, using edge computing, the present research proposes obtaining small data stored inside the manufacturing system and using it to implement HCPS. Thus, we argue that by optimizing big data, it is possible to prepare a smaller data set, containing precisely selected data samples that are value-adding for further use in smart quality management. A smaller data set that is created in this way can be defined as a precisely selected small data set obtained from big data via optimization without losing significant information contained in the big data in the first place. Not losing significant information means that the use of small data will not reduce the quality of the system's self-prediction capabilities since it contains only value-adding data samples. Therefore, the present research develops a new conceptual model for edge computing data optimization by creating small data which enable timely, meaningful insights into manufacturing systems, and are organized in an accessible and understandable way, thus optimizing resource efficiency and contributing to the sustainability of manufacturing systems without reducing the quality of the system's self-prediction capabilities. The conceptual model—inspired by data mining methodology within the Industry 5.0 concept emphasizing the human knowledge aspect, resilience empowered by edge computing, and sustainability through resource optimization—consists of three main phases divided into several steps. Finally, a proof-of-concept is done using manufacturing data from a process industry company.

The present research is organized as follows. Section 2 provides a theoretical background of Industry 5.0, manufacturing data, and edge computing. Section 3 presents the

research method and provides details of the conceptual model development and validation. Section 4 presents the results of the conceptual model development as well as the proof of concept using the manufacturing data. Finally, Section 5 discusses the results, sums up the contributions, and derives conclusions.

## 2. Background

### 2.1. Smart Quality Management in Industry 5.0

In recent years, the focus on smart manufacturing systems has been pushing industry toward a new variety of highly advanced technical solutions. In fact, smart manufacturing systems often incorporate smart quality management [29] optimization capabilities to reduce time and cost for improvements of the entire production efficiency using a technology-oriented approach, such as Industry 4.0. However, Industry 5.0 proposes a new phase of industrial development that builds on the previous phases of industrialization, with a focus on digital sustainability and human-centred approaches [30] to technology. On the one hand, Industry 5.0 has a strong connection to digital sustainability [31]. One key aspect of Industry 5.0 is the integration of advanced digital technologies into manufacturing processes. These technologies can be used to optimize production processes, reduce waste and energy consumption, and improve overall efficiency. On the other hand, the European Commission pointed out that humans are still the most important asset of every company: they are dexterous, intelligent, flexible, and creative, and outperform most machines or robots (European Commission 2019). In reply to this focus change, the smart quality management in the Industry 5.0 concept was introduced emphasizing the main role of the research and innovation sector in support of industry in its long-term service to humanity [5].

Therefore, it is important to accent that smart quality management creation in Industry 5.0 is not an alternative nor chronological continuation of the present Industry 4.0 concept [7]. It is the result of a forward-looking perspective for the need of engineers' and workers' skills, knowledge, and abilities to cooperate with machines and robots on the one side, and flexibilities in manufacturing processes and environmental impacts on the other. Thus, smart quality management in Industry 5.0 complements and extends Industry 4.0 [5,7].

Since Industry 5.0 is a new concept, the official definitions still represent abstract ideas generalised from practice, focusing primarily on human aspects [11,32] and sustainable and resilient intelligent production [5,33]. As such, early definitions of Industry 5.0 depend on the research field. Consequently, we decided to provide a definition of Industry 5.0 that focuses on smart quality management, with an emphasis on a human-centric industry environment. Notably, the definition of Industry 5.0, derived from the cited references, reads as follows:

Industry 5.0 represents the concept of transition to a human–centric, sustainable, and resilient industry [5,9,34] to improve process, products, and systems quality, driven by advanced technologies, grouped into categories (adapted from [35]) for:

- individualised human–machine interaction (including artificial intelligence robotics, and augmented and virtual reality);
- manufacturing system simulation (including CPS, digital twins of products, processes, and entire systems); and
- data transmission, storage, and analysis technologies (including IIoT, cyber-security, big data analytics, and edge computing).

### 2.2. Manufacturing Data

Big data in manufacturing (hereinafter big data) refers to shop floor data collected during the manufacturing process. Big data implies a massive amount of raw, highly distributed structured and unstructured data generated by multisource sensors [21,22], intersystem communication, and related external information that is transformed into value using new technology and analytical methods [19,23]. Thus, big data transformed into valuable information provides new solutions to improve manufacturing system productiv-

ity [21,24], increase productivity through quality improvement [30], and enable predictive and proactive maintenance using manufacturing data [36].

However, the generation of big data also brings challenges [8] (e.g., massive amounts of data to manage, store, and process; insufficient quality of the collected data; insufficient data processing power [6], etc.), where the amount of relevant data per parameter is small, and hence, it may lead to imprecise estimations [37]. Therefore, it is necessary to systematically reduce big data to precisely selected small data sets. Small data in manufacturing (hereinafter small data) is the preprocessed and optimised information obtained from big data [38] using computing technologies that consume less power and enable real-time processing and efficient decision-making using advanced analytical methods. Therefore, data preprocessing and optimization emphasise the role of edge computing as a technology that can be utilised for data reduction [39]. Thus, we argue that by optimizing big data, and using edge computing as a more efficient and sustainable technology compared to cloud computing, it is possible to prepare a small data set for use in smart quality management. These small data enable timely, meaningful insights into the system, organised in an accessible and understandable way.

### 2.3. Edge Computing

Edge computing transforms the way data are handled, processed, and used from various manufacturing sources (i.e., machines and devices) [39]. Recently, industry has more frequently opted for edge computing technology [40]. This choice is driven by the growth of IIoT-connected manufacturing resources, the application of data analytical methods that require computing technology that consumes less power, and the need for real-time decision-making [27,39]. This trend is also recorded in the increasing number of scientific papers dealing with the application of edge computing in manufacturing systems [41,42]. The reason for this trend is that edge computing's distributed computing allows data processing and storage close to manufacturing resources, which supports sustainable and resilient production [26].

However, edge computing has limitations regarding power and processing resource utilization and storage space size [27,28]. For this reason, working with small-scale data sets [43] results in the optimised use of resources for power and processing, while reducing the required storage space and analysis costs [43]. Thus, there is a need for both industry and academia to substitute big data with precisely selected small data sets for use in smart quality management systems without the loss of significant big data information [14].

## 3. Research Method

According to Phaal et al. [44], the term 'conceptual' implies 'concerned with the abstraction or understanding of a situation'. Jose et al. [45] describes modelling and reasoning on models as a 'basic human capabilities for coping with, understanding, and influencing the environment'. Conceptual modelling, as a research activity, 'formally describes some aspects of the physical and social world around us for purposes of understanding and communication' [46]. Therefore, conceptual modelling is one of the kernel activities in information systems engineering [45].

The present research applies conceptual modelling [47–49] to develop a model for edge computing data optimization in Industry 5.0. The developed model was inspired by human-cyber-physical integration for data mining in the framework of the Industry 5.0 ecosystem [50–52]. Notably, the model development was informed by the practical field experience of the research team in the implementation of smart quality management systems in industry.

Subsequently, a proof-of-concept method [53,54] is provided for the developed model for edge computing data optimization by applying it in industry using manufacturing data on product quality. According to [55], a proof-of-concept, as a research practice, serves as an instrument of knowledge construction in the study which has a set of activities (i.e., actions, movements, analyses, simulations, techniques, and tests, among others) for the

assessment, understanding, validation, exploration, and learning of developed models in a given area of knowledge.

Specifically, using a proof-of-concept method, the developed model was tested in real industry conditions on manufacturing data collected in the process industry—that is, on the production line of a vinyl floors company. The equipment used for the proof-of-concept was the MELIPC MI5000 industrial computer developed by Mitsubishi Electric. The MELIPC MI5000 is based on edge computing for real-time data collection, analysis, diagnosis, and feedback, and as such is fit to validate the developed conceptual model using real industry data.

For the realization of the proof-of-concept, the research team, company engineers, and shop floor workers cooperated. The research team consisted of 5 scientists from a multidisciplinary field of industrial engineering and computer science. The expert team from the company provided 2 people from top management, 3 senior engineers, and 5 workers from the production. That is in total 15 experts. The realization of the proof-of-concept lasted six months—covering the period from data collection to the final optimization of the big data set into a precisely selected small data set—within which interview meetings were held in the intervals of every two weeks.

## 4. Results

### 4.1. Conceptual Model Development

In the present subsection, the developed edge computing data optimization conceptual model (Figure 1) is presented and explained. The model is composed of three phases: Phase 1: smart quality management problem definition; Phase 2: parameter identification and IIoT data collection; and Phase 3: edge computing data preprocessing. The details of the phases and steps of the model are presented in Table 1 to avoid redundancy and facilitate the reading of the paper. Data optimization for smart quality management empowered by edge computing from an Industry 5.0 perspective encompasses the developed conceptual model, which is presented in Figure 2.

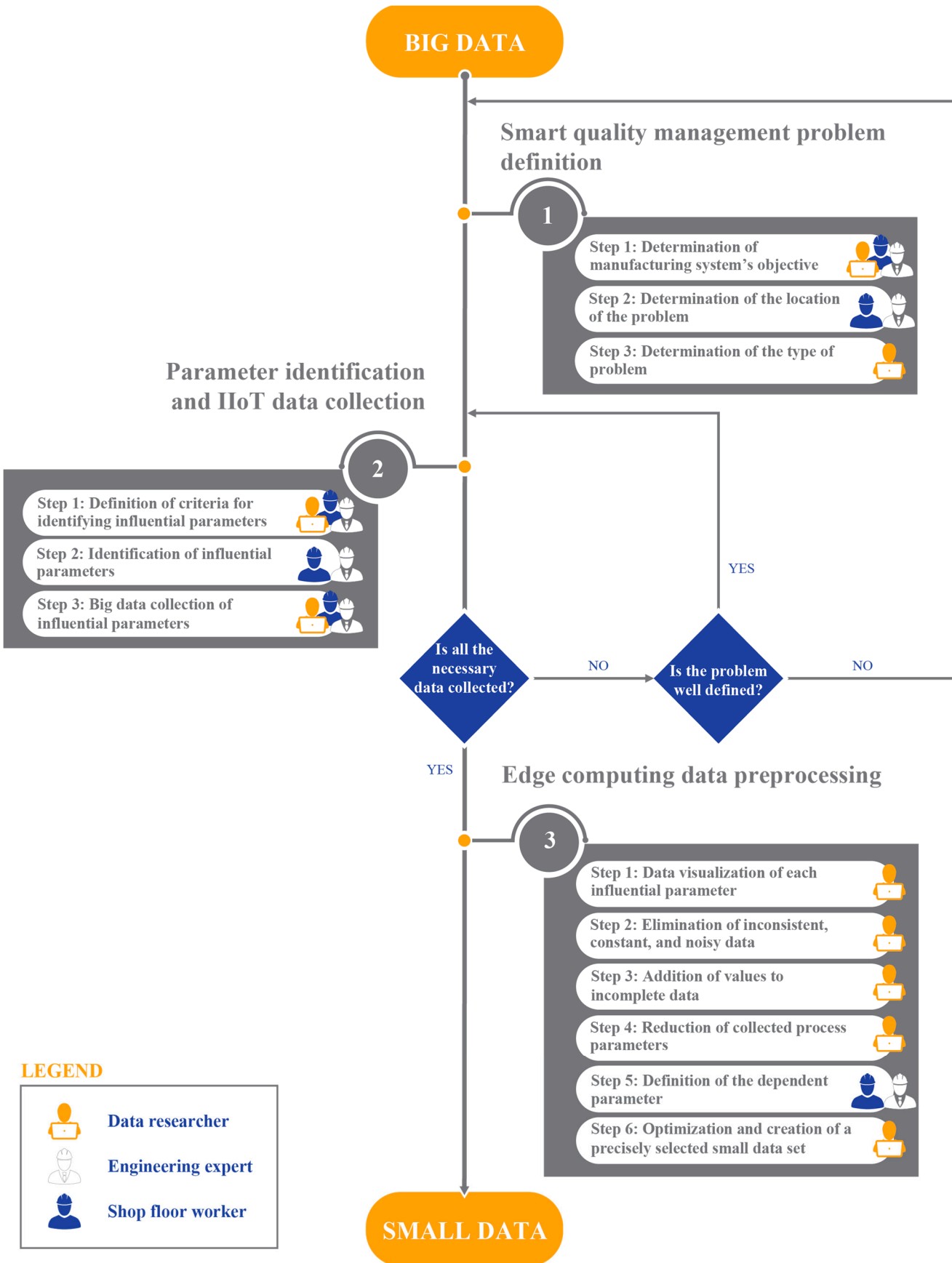

**Figure 1.** The conceptual model for edge computing data optimization for smart quality management.

**Table 1.** Phases and steps of the developed edge computing data optimization conceptual model.

| Phase | Step | Step Definition | Step Execution Method | Participants | | |
|---|---|---|---|---|---|---|
| | | | | Data Researchers | Engineering Experts | Shop Floor Workers |
| 1—Smart quality management problem definition | 1. Determination of manufacturing system's objective | Establishment of the manufacturing system's objective based on the established business goal of the company | Interviews with engineering experts and shop floor workers | X | X | X |
| | 2. Determination of the location of the problem | Consideration of all potential locations of the problem (with the possibility that the problem location may include one or more machines in the manufacturing system) | Experts' and workers' analysis | | X | X |
| | 3. Determination of the type of problem | Establishment of the data mining goal based on the company's business goal, with an emphasis on data mining methodology (data mining problem type, namely, classification or regression) | Data mining | X | | |
| 2—Parameter identification and IIoT data collection | 1. Definition of criteria for identifying influential parameters | Definition of the criteria for process parameter identification based on the company's business goal | Interviews with engineering experts and shop floor workers | X | X | X |
| | 2. Identification of influential parameters | Selection of the influential process parameters on the basis of defined criteria | Experts' and workers' analysis | | X | X |
| | 3. Big data collection of influential parameters | Collection of data for identified process parameters in a standardized format (TXT, XLS, CSV, JSON, XML, etc.) in real–time for a defined period of time via IIoT network | Edge computing | X | X | X |
| 3—Edge computing data preprocessing | 1. Data visualization of each influential parameter | Visual presentation of the collected data for influential process parameters that provides information about their nature (significant deviations exist [Figure 3a]) or no significant deviations [Figure 3b]) from the usual values of process parameters generated during the manufacturing process) | Visual analysis | X | | |
| | 2. Elimination of inconsistent, constant, and noisy data | Manual removal of data that are considered to have significant defects (constants or noisy values of data due to malfunctions in the measuring instruments) | Manual rows deletion—timestamp data OR Manual columns deletion—numerical or categorical data Note: The method is chosen depending on the type of data | X | | |

**Table 1.** *Cont.*

| Phase | Step | Step Definition | Step Execution Method | Participants | | |
| --- | --- | --- | --- | --- | --- | --- |
| | | | | Data Researchers | Engineering Experts | Shop Floor Workers |
| | 3. Addition of values to incomplete data | Data insertion of missing values in the empty spaces in a two–dimensional matrix by applying appropriate analytical methods for value addition based on data type | Mean/median/mode— missing values of numeric data OR Multiple imputation—missing values of numerical or categorical data OR Seasonal adjustment + linear interpolation— timestamp data with both trend and seasonality Note: The method is chosen depending on the type of data | X | | |
| | 4. Reduction of collected process parameters | Reduction of data dimensionality using correlation analysis or multiple linear regression to decrease the number of independent process parameters and simplify the data set | Correlation analysis— classification data mining type OR Multiple linear regression— regression data mining type Note: The method is chosen depending on the data mining type | X | | |
| | 5. Definition of the dependent parameter | Selection of the output value of process parameters based on the established data mining goal | Experts' and workers' analysis | | X | X |
| | 6. Optimization and creation of a precisely selected small data set | Preprocessing of the collected data set consists of influential process parameters using range analysis to calculate the difference between the maximum and minimum values by reducing all collected files to one file with the same dimension. This is followed by linking the preprocessed data set to the dependent parameter | Range analysis | X | | |

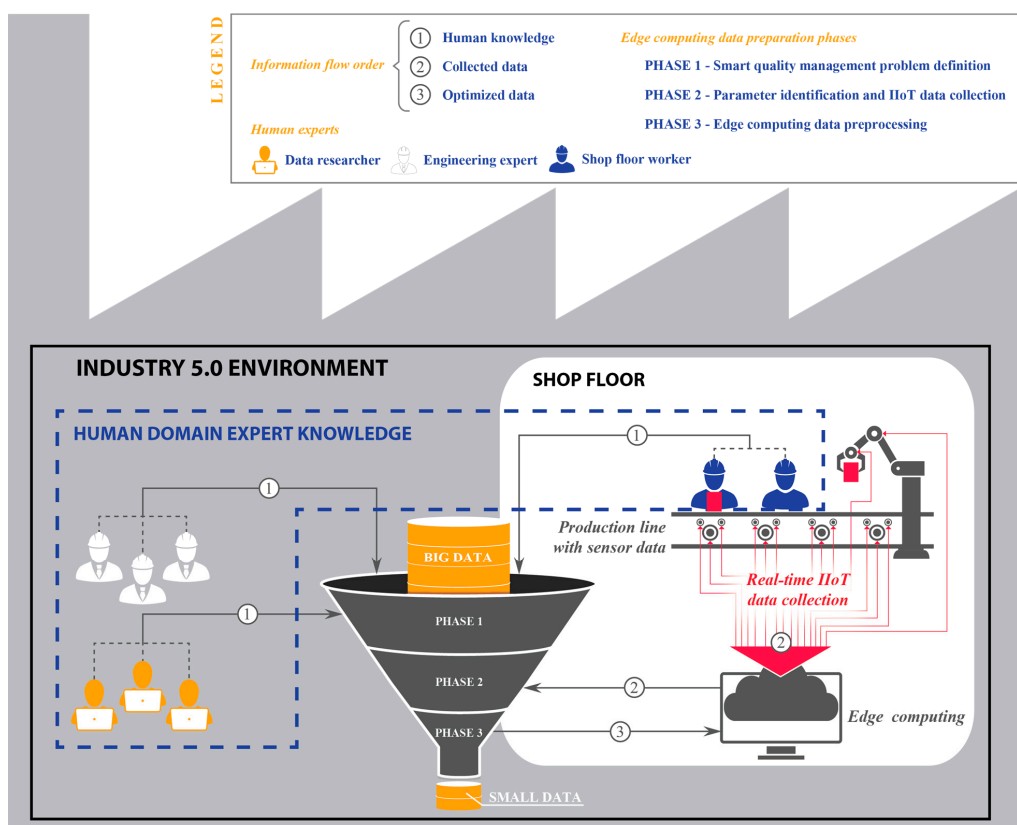

**Figure 2.** Data optimization for smart quality management empowered by edge computing from Industry 5.0 perspective.

### 4.1.1. Phase 1—Smart Quality Management Problem Definition

Phase 1 aims to define the problem in a systematic way for the specific manufacturing system. This implies gaining an adequate understanding of the processes and activities within that manufacturing system. Specifically, in order to transform a manufacturing system into a smart manufacturing system in Industry 5.0, it is essential that data researchers fully understand how the particular processes work in the company, which will enable them to gain an awareness of which data should be analysed and modelled and which advanced analytical method should be used. Achieving a proper understanding of an entire manufacturing system is a complex task due to the necessity of obtaining knowledge about all of the processes as well as establishing cooperation with the engineering experts and shop floor workers who provide that knowledge to data researchers. The developed conceptual model proposes three steps for the smart quality management problem definition (Phase 1): (1) determination of the manufacturing system's objectives; (2) determination of the location of the problem; and (3) determination of the type of problem (for details, see Table 1).

### 4.1.2. Phase 2—Parameter Identification and IIoT Data Collection

This phase enables the real-time collection of data generated during the manufacturing process. Notably, it is necessary to ensure systematic data collection. This involves the process of constant communication and discussion between engineering experts, shop floor workers, and data researchers. Subsequently, the data researchers prepare the data based on the knowledge and experience of experts and workers. Notably, when using edge computing data optimization in Industry 5.0, it is necessary to ensure the optimal use of the available database space. The collection of data related exclusively to the identified influential process parameters, which directly affect the occurrence of a defined problem in the manufacturing system, enables the optimal use of database space. The developed conceptual model proposes three steps for parameter identification and IIoT data collection

(Phase 2): (1) definition of criteria for identifying influential parameters; (2) identification of influential parameters; and (3) big data collection of influential parameters (see Table 1) that are visualized to determine the quality of the collected data (Figure 3).

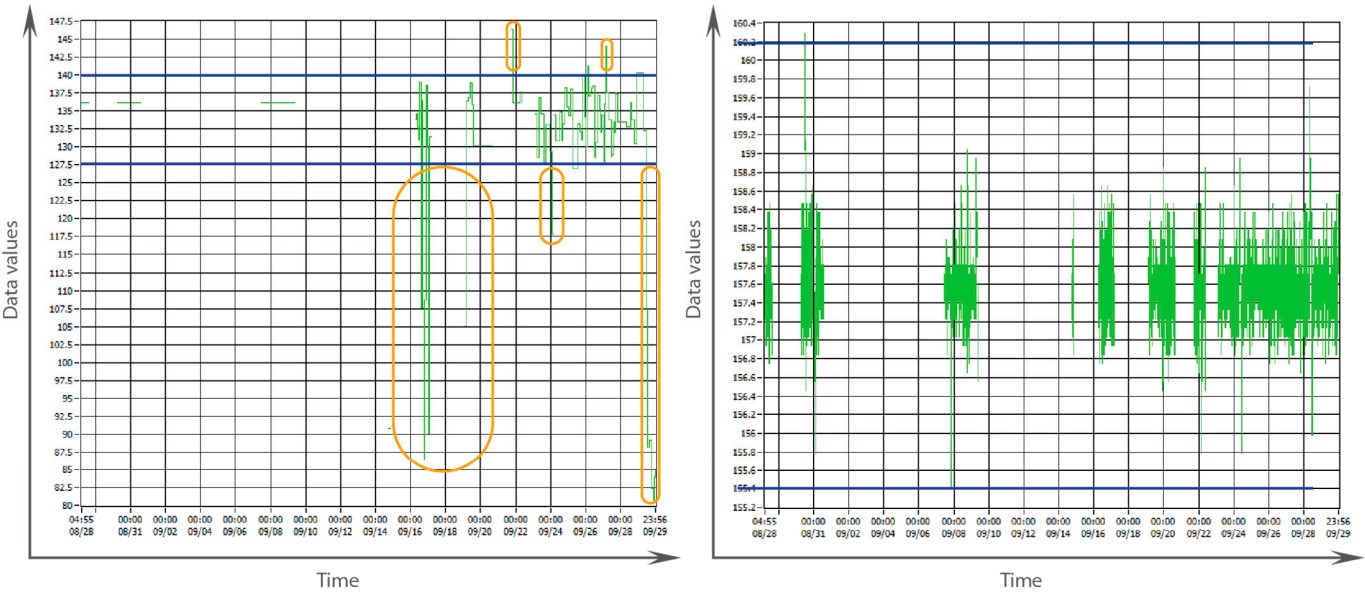

NOTE: The discontinuity in the visualization of data implies that representative product group was not produced at that time

LEGEND: —— defined tolerances
—— significant deviations

(**a**)          (**b**)

**Figure 3.** Visual presentation of influential process parameters collected in the case company: (**a**) Significant deviations exist (data values are not within the defined tolerances); (**b**) No significant deviations exist (data values are within the defined tolerances).

### 4.1.3. Phase 3—Edge Computing Data Preprocessing

The edge computing data preprocessing phase covers all activities to construct the final optimised small data set prepared to feed into the mathematical data modelling tools in order to perform an analysis of the process data and develop a mathematical model for data processing. Tasks include table records analysis and attribute selection, as well as the transformation and cleaning of data for the modelling tools. Therefore, the starting point of Phase 3 is an adequate understanding of the collected raw data (Phase 2). Understanding the raw data sets means interpreting the values of the collected big data. The developed conceptual model proposes six steps for preprocessing the edge computing data (Phase 3): (1) data visualization of each influential parameter, (2) elimination of inconsistent, constant and noisy data, (3) addition of values to incomplete data, (4) reduction of collected process parameters, (5) definition of the dependent parameter, and (6) optimization and creation of a precisely selected small data set. Notably, it is not necessary to implement all the defined steps from the data preprocessing phase when applying the model in a manufacturing environment. The implementation of the defined steps is reflected in the generalizability of the developed conceptual model, which depends exclusively on the nature of the collected big data and the problem defined in Phase 1 (see Table 1).

### 4.2. Proof-of-Concept

This section presents the proof-of-concept for the developed edge computing data optimization conceptual model. The proof-of-concept was accomplished by applying the conceptual model to the timestamp manufacturing data collected from an industry in an Industry 5.0 human-cyber-physical environment (Figure 2). The main goal of the

case company in implementing the developed conceptual model was the detection of poor-quality products in production to increase the company's efficiency.

Specifically, we applied the developed conceptual model to manufacturing data collected from the vinyl flooring industry. Vinyl flooring is produced and delivered in the form of a roll, which conditions the existence of a continuous production line. The specific production line consists of 12 machines and is 850 m long. Since the number of machines increases the complexity of the manufacturing process and planning and control for production, the operations divided the production line into 5 clusters. A cluster is a part of the production line that includes one or more machines, where experience has shown that inconsistencies in the process parameters on one machine can potentially result in a poor-quality product on another subsequent machine. In addition, the machines must be grouped in the sequence of production operations.

The equipment used for validation of the developed conceptual model is an industrial computer based on edge computing for real-time data collection and analysis. The applied edge computing technology is the MELIPC MI5000, a solution developed by Mitsubishi Electric. The goal of using the MELIPC MI5000 is to systematically reduce big data by optimizing it into a precisely selected small data set ready to be used for later data modelling. The results of the proven concept based on the manufacturing data are presented in Table 2.

**Table 2.** Proof-of-concept results on the data from the case manufacturing system.

| Phase | Step | Step Execution Method | Results |
|---|---|---|---|
| 1—Smart quality management problem definition | 1. Determination of manufacturing system's objective | Interviews with engineering experts and shop floor workers | • Defined manufacturing system's objective: Improved quality of the selected product (from the representative group) for reducing the number of products that do not meet the required quality level |
| | 2. Determination of the location of the problem | Experts' and workers' analysis | • Defined location of the problem: the second coating machine, Cluster 1 |
| | 3. Determination of the type of problem | Data mining | • Defined problem type: classification |
| 2—Parameter identification and IIoT data collection | 1. Definition of criteria for identifying influential parameters | Interviews with engineering experts and shop floor workers | • The defined criterion: Selection of the process parameters based on similar tolerances for the representative group of products (the representative group of products is the one produced in the largest quantities in the manufacturing system when looking at annual production) |
| | 2. Identification of influential parameters | Experts' and workers' analysis | • Number of influential parameters identified: 15<br>• (line speed, drum temperature, viscosity, etc.) |
| | 3. Big data collection of influential parameters | Edge computing | • Data collection period: 33 days<br>• Data format: CSV file<br>• Number of collected files: 6534<br>• Number of data in each file: 300<br>• Total amount of data units: 29,403,000 |

**Table 2.** *Cont.*

| Phase | Step | Step Execution Method | Results |
|---|---|---|---|
| 3—Edge computing data preprocessing | 1. Data visualization of each influential parameter | Visual analysis | • Unbalanced data set |
| | 2. Elimination of inconsistent, constant, and noisy data | Manual rows deletion of timestamp data | • Number of reduced files: 3802<br>• Total amount of data units: 17,109,000 |
| | 3. Addition of values to incomplete data | Visual analysis | • Not implemented due to non–existence of incomplete data values |
| | 4. Reduction of collected process parameters | Correlation analysis | • Number of remaining influential parameters after reduction: 12<br>• Total amount of data units: 13,687,200 |
| | 5. Definition of the dependent parameter | Experts' and workers' analysis | • The dependent parameter: binary (a value of '0' indicates that the final product has no defects, while a value of '1' indicates that the final product has defects in the form of poor quality.)<br>• Total number of poor-quality final products: 5 |
| | 6. Optimization and creation of a precisely selected small data set precisely selected | Range analysis | • 3802 CSV files into a single unique CSV file containing 3802 data samples<br>• Total amount of data units: 45,624 |

Note: The third column is repeated from Table 1. Although redundant, it is repeated to enable readers to better follow the results exposition.

*Phase 1—Smart quality management problem definition* (Table 2): The first phase of the conceptual model was one of the most demanding for the verification of the conceptual model. The reason for this is the difficulty of defining the problem when the manufacturing process is complex. The complexity of the manufacturing process is reflected in the interdependence of all process parameters for the representative group of products (the representative group of products was composed of products that share the same parameter configuration for the observed cluster of machines—Cluster 1) on the entire production line. The interdependence of all process parameters means that inconsistencies and non-compliances in process parameters that occur, for example, at the beginning of the production line, lead to the appearance of the observed poor-quality product in one of the subsequent operations. This leads to inaccuracies when deducing the exact location of a particular discrepancy. In very complex manufacturing processes, it is difficult to consider the entire production line to locate where a problem occurs. As a result, through expert analysis, Cluster 1 was identified as having the greatest possibility that a mismatch in the process parameters on one machine could potentially lead to poor-quality products on one of the subsequent machines. Furthermore, the analysis showed that preventing poor-quality products from exiting Cluster 1 would lead to significant waste reduction on the whole process line. Notably, Cluster 1 consists of three machines: first coating, second coating, and printing. Additionally, by analysing the production data, it is found that the highest number of poor-quality products in the previous production period was detected in Cluster 1, specifically on the second coating machine. Thus, it was found that the interdependence of process parameters on the selected machine is minimised, since the machine is located at the very beginning of the production line where the emergence of poor-quality products is exclusively related to Cluster 1. Therefore, the location of the problem in the case company is the second coating machine, based on which the smart quality management problem is defined as a classification type in data mining.

*Phase 2—Parameter identification and IIoT data collection* (Table 2): In this phase, the influential process parameters were defined. Fifteen influential process parameters were identified for the representative group of products on the second coating machine. Then, data for the influential process parameters were collected in real-time, enabled by the application of edge computing. Edge computing was used in the manufacturing system, where data were collected at the point in the manufacturing process where they were generated. This enabled a later analysis of the process data from the time of their collection. The application of edge computing also required the development of data modelling on an optimised, precisely selected small data set. Therefore, a 33-day time period for collecting process parameters was chosen for two reasons, namely due to the storage limitations of edge computing technology and because most of the products belonging to the relevant product group were produced in that time period (according to the production plan and program). In this phase, using edge computing for the defined time period, 29,403,000 data units were collected. The quantity of data collected indicates that the limitations of the edge computing storage space did not negatively impact the number of collected data units.

*Phase 3—Edge computing data preprocessing* (Table 2): This phase was performed using edge computing technology. Based on the visual presentation of the data, it was found that most of the process parameter values are within the defined tolerances for process parameters. Therefore, the collected process parameter values represent an unbalanced set of data that mostly contains data on the production of final class 'A' products (i.e., 'good' data) and a very small number of 'bad' data that provide information about poor or insufficient product quality. By further monitoring and elaborating the steps in this phase, the total number of data units collected decreased significantly. The reasons for this are as follows:

- elimination of inconsistent, constant, and noise data was done based on expert knowledge which included the manual rows deletion of timestamp data. That led to a reduction in the total number of data units collected by approximately 50%, leaving a total amount of 17,109,000 data units (see Figure 4 and Table 2); and
- reduction of the number of collected process parameters using correlation analysis, which led to a decrease in the total number of influential process parameters, from 15 to 12, based on the Pearson correlation coefficient [56,57] (Table 2), by ejecting parameters that are highly correlated ($\rho$ exceeds the value of 0.8 and $-0.8$, respectively, where values are calculated based on Equation (1) from Figure 4).

Subsequently, a dependent parameter of binary character is defined, which provides information when quality errors occur in the collected data set. The binary character of the dependent parameter was chosen for easier product classification. Further, to determine the variability of the parameters using range analysis [58,59], each individual CSV file was analysed, after which the value of the difference between the maximum and minimum values was obtained. In other words, range analysis was used to optimise 3802 CSV files into a single unique CSV file (Figure 4). Notably, this unique CSV file contains information about all 3802 CSV files. Thus, the total amount of data was reduced to 45,624 data units (Figure 4). After optimization, the values of the collected process parameters and the values of the dependent parameter were manually paired to create a precisely selected small set of data for optimizing edge computing data for smart quality management from the Industry 5.0 perspective.

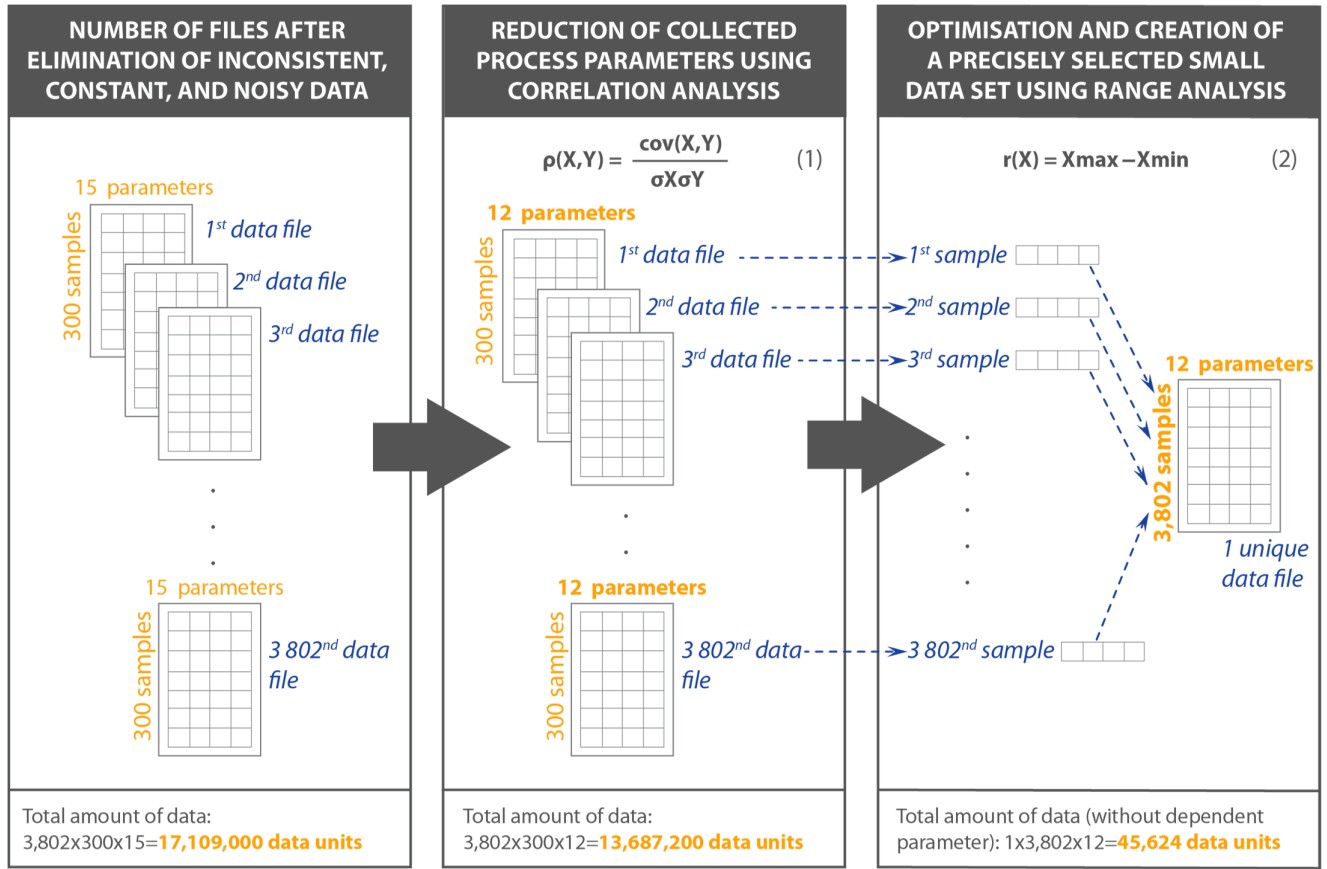

NOTE:

Equation (1) represents the calculation for the Pearson correlation coefficient ($\rho$) in correlation analysis based on which the parameter reduction was performed.

Equation (2) represents the range calculation (r) of each process parameter for the reduction of every data file into a single sample, with the aim of creating a single unique data file withouth losing significant information.

**Figure 4.** Graphical presentation of proof-of-concept for the edge computing data preprocessing phase of performed steps 2, 4, and 6 for optimization of the collected data files without losing significant information.

## 5. Discussion and Conclusions

In the present research, innovative big data optimization is proposed using edge computing inspired by human-cyber-physical integration for data mining in the Industry 5.0 ecosystem. Recently, researchers have started to point out the advantages of small data [60,61], characterised as 'the tiny clues that uncover huge trends' [62]. We argue that optimised small data (sets) enable timely, meaningful insights into the manufacturing system that are organised in an accessible and understandable way, while enabling real-time decision-making. Thus, the present research showed that starting from and optimizing big data, and by using edge computing technology, it is possible to prepare a precisely selected small data set for use in smart quality management without losing significant information contained in the big data. In the following paragraphs, the main contributions of the present research are presented.

*The proposed conceptual model is an innovative big data optimization method*. The proposed conceptual model was developed as an innovative big data optimization method for edge computing aligned with real-time data acquisition and smart quality management problem definition, and inspired by the industrial data mining approach. The developed model consists of three main phases (Figure 1): smart quality management problem definition, parameter identification and IIoT data collection, and edge computing data preprocessing. Each phase is divided into a number of steps defined for the adequate and unambiguous implementation of the model. The development of the conceptual model was motivated by the lack of an adequate methodology for preprocessing manufacturing data in the

literature. At the same time, the research [63] stresses the importance of data preprocessing because: (1) real-world data is impure; (2) high-performance mining systems require quality data; and (3) quality data yields concentrative patterns. Moreover, many preprocessing techniques have been reviewed in the scientific literature [64–66], but the lack of data preprocessing methodologies still remains. Further, in the reviewed scientific literature, the cross-industry standard process for data mining (CRISP–DM) [67,68] methodology, in which researchers focus on big data [69–71], is most often used as a basis for data preprocessing [72,73]. However, the need and possible challenges for the optimization of big data collected in manufacturing systems into a small data set are not considered in CRISP-DM. Thus, the present conceptual model using edge computing was developed with the aim of the systematic reduction and optimization of big data into a small and precisely selected data set ready to be used for the modelling, testing, and subsequent deployment of the developed predictive model. By fulfilling this aim, we have successfully responded to the gap in the data preprocessing methodologies in the available literature.

The proposed conceptual model promotes the knowledge integration of blue-collar workers (shop floor workers) and white-collar workers (engineering experts) with data researchers. Human knowledge plays the central role in the developed conceptual model, as it should in manufacturing systems [5,9]. Knowledge integration is obtained in the synergy of industry and academia, where human domain expert knowledge, along with Industry 5.0 advanced technologies (i.e., IIoT, edge computing, and advanced analytics methods), is the basis for the creation of quality management systems [29]. Informed by the conducted proof-of-concept, it is expected that the implementation of the developed conceptual model in industry will: (1) reduce the cost for big data storage, since a much smaller data set can be used; (2) achieve more secure processes with the use of edge computing; and (3) enable the inclusion of human domain expert knowledge to create smart quality management systems.

*The developed conceptual model enables the implementation of smart quality management from the Industry 5.0 perspective.* The developed conceptual model points out the central position of the human factor [5,9] in smart quality management systems. The human domain expert knowledge, physical, and virtual components are integrated via the developed model (Figure 2). This integration is enabled by the application of the Industry 5.0 human-centered concept and advanced technologies [35]—for example the IIoT, edge computing, and advanced analytics methods—via the developed conceptual model. Further, the proof-of-concept performed with manufacturing data validated the future possibility of creating HCPS in the case of a vinyl floor producer. Therefore, the proposed methodology (i.e., the developed conceptual model) supports the development of HCPS and contributes to digital sustainability and further research on the Industry 5.0 concept (Figure 2).

*The proof-of-concept was performed with the use of real industry data.* The developed conceptual model was tested based on industrial application through manufacturing data optimization in a company from the vinyl flooring sector. At the time of this study, there is a gap of practical proof in the relevant literature on the developed theoretical frameworks for both Industry 4.0 and Industry 5.0 [6,74–76]. Specifically, evidence of the implementation of data analytics, such as Industry 4.0 and Industry 5.0 technology, is scarce [68,77–79]. To address this gap, the present research demonstrated that by using the developed conceptual model for edge computing data optimization for smart quality management from the Industry 5.0 perspective, it is possible to prepare a precisely selected small data set without losing significant information contained in the big data. Notably, the manufacturing system's objectives were considered in the developed conceptual model when optimizing the manufacturing data.

*The proof-of-concept was performed using manufacturing small data.* The conceptual model was validated by proving the concept with manufacturing data from the case company that was optimised into small data [38,80,81]. The proof-of-concept was achieved by applying the optimisation approach of the proposed conceptual model (Figure 1) based on expert domain knowledge (Figure 2). Notably, the manufacturing problem was defined, influential

parameters were identified, and data were collected using the IIoT network. As a result, by using edge computing data optimization, the 29,403,000 data units (i.e., big data) were reduced to 45,624 data units (i.e., small data) without losing the information contained in the big data for decision-making. The preservation of information was achieved by decreasing the number of influential process parameters by ejecting parameters that were highly correlated via correlation analysis, and reducing collected data files into single samples for each file via range analysis (see Figure 4 and Table 2) [60,61]. Then, samples were merged into a unique file containing 3802 samples for 12 influential process parameters (Figure 4). Notably, the number of data units was reduced by 99.73%.

We are aware that the development of a conceptual model for edge computing data optimization opens additional research opportunities in the field of smart quality management, given that the optimised small data set needs to be further analysed and processed. It further opens the question of whether optimised small data would be sufficient to achieve high accuracy in the remaining data mining phases (i.e., modelling the data, training and testing, and verification and deployment). These questions, however, are beyond the scope of the present research. In this research phase, the data set for edge computing was optimised according to state-of-the-art statistics methods, and prepared for further data mining considering the most important aspects of Industry 5.0; i.e., digitally sustainable, human-centric and resilient manufacturing processes. Future research will focus on the remaining data mining phases based on optimised small data sets for further advancement of edge computing self-prediction applications from the Industry 5.0 perspective.

**Author Contributions:** Methodology, B.B., S.M., M.S., M.J. and A.R.; Software, B.B.; Validation, N.S., M.S., M.J. and A.R.; Formal analysis, B.B., N.S. and A.R.; Investigation, B.B. All authors have read and agreed to the published version of the manuscript.

**Funding:** This research received no external funding.

**Institutional Review Board Statement:** Not applicable.

**Informed Consent Statement:** Not applicable.

**Data Availability Statement:** The data are not publicly available due to privacy restrictions.

**Acknowledgments:** This paper has been published as a part of the project that is financed by the Science Fund of the Republic of Serbia within its program "IDEAS"—Management of New Security Risks—Research and Simulation Development, NEWSIMR&D, #7749151.

**Conflicts of Interest:** The authors declare no conflict of interest.

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
