# Peer review of "Edge Computing Data Optimization for Smart Quality Management: Industry 5.0 Perspective"

_sustainability, doi:10.3390/su15076032_

Round 1
Reviewer 1 Report
The paper under review claims to develop a conceptual model to promote Industry 5.0. The aim of the developed model is to optimize big data to fewer data without losing significant information contained in big data. The model is empowered by edge computing, as an Industry 5.0 enabler. In this way, we aim to optimize data storage and create conditions for further power and processing resource rationalization. Finally, an industrial case study was applied through a proof-of-concept using real manufacturing data from a vinyl flooring industry, where the amount of data was reduced.
- This research paper includes several claims that were not proven in the context of the paper; the authors claimed that they created a conceptual model, the main objective of a conceptual data model is to define what the system contains clearly to define business concepts and rules. What I found here is not a clear and well defined Quality Management model.
- How does the proposed model contribute to promoting Industry 5.0
- The method of optimizing data storage and resources should be the core of this research and should be well explained.
- There is nothing to prove or validate that the amount of data was reduced by 99.85%
- Digital Sustainability was not addressed sufficiently to be related to the Journal
- All figures need to be revised, they are not clear, not illustrative, the notations of the axes in figure 3 are missing, figure 4 is a mess and it does not show the five clusters of the production line
- Excessive statistical analysis of the outputs of the clusters should be presented to verify the clusters with quality problems
- The correlation technique used to eliminate inconsistent data should be presented
- The English lanquage should be revised, "a" and "an' are used extensively (ex.: page 3,5,7, ....), spaces between several words are missing (page 3: powerand-14, page 4: conceptmethod-19, conceptwas-19, page 11: parametersmeans-31,.....)
- The authors used "we" and "our" several times
Reviewer 2 Report
Abstract
Please add briefly contribution of your study to knowledge and practices
Introduction:
Citation space issue line 43, 49 please check for all page
73 74, avoid "we" please revise
You said in abstract
However, it seems Industry 4.0 hype did not fulfil industry expectations due to many implementation challenges. Please describe it in intro duction more detail
Background section
Page 124-132, change italic to normal style
Research Method
Please make a flow chart of your research, seems not clear only by text
How you determine 15 expert? page 206, please described
Result:
Figure 4 is not clear and difficult to understand, please revise
Discussion:
Need to discuss the limitation Industry 4.0 and 5.0 deeper
Reviewer 3 Report
The authors represent the model of optimization of an industrial big data without losing significant information contained in big data.
The work is interesting and easy for reading but I would recommend reduse a size of the section "Discussion and Conclusions" by moving part of content to "Results".
Round 2
Reviewer 1 Report
The authors were not fully able to provide adequate answers to the mentioned comments in the previous report
Reviewer 2 Report
Thank you for addressing my comment, now seem much better
